# Establishment of a Gene Signature to Predict Prognosis for Patients with Lung Adenocarcinoma

**DOI:** 10.3390/ijms21228479

**Published:** 2020-11-11

**Authors:** Zhaodong Li, Fangyuan Qi, Fan Li

**Affiliations:** 1The Key Laboratory of Zoonosis, Department of Pathogenobiology, College of Basic Medicine, Jilin University, Changchun 130021, China; lizd17@mails.jlu.edu.cn (Z.L.); qify19@mails.jlu.edu.cn (F.Q.); 2The Key Laboratory for Bionics Engineering, Jilin University, Changchun 130021, China; 3Engineering Research Center for Medical Biomaterials of Jilin Province, Jilin University, Changchun 130021, China; 4Key Laboratory for Biomedical Materials of Jilin Province, Jilin University, Changchun 130021, China; 5State Key Laboratory of Pathogenesis, Prevention and Treatment of HighIncidence Diseases in Central Asia, Urumqi 830000, China

**Keywords:** lung adenocarcinoma, gene signature, overall survival, prognosis, bioinformatic analysis

## Abstract

Accumulating evidence indicates that the reliable gene signature may serve as an independent prognosis factor for lung adenocarcinoma (LUAD) diagnosis. Here, we sought to identify a risk score signature for survival prediction of LUAD patients. In the Gene Expression Omnibus (GEO) database, GSE18842, GSE75037, GSE101929, and GSE19188 mRNA expression profiles were downloaded to screen differentially expressed genes (DEGs), which were used to establish a protein-protein interaction network and perform clustering module analysis. Univariate and multivariate proportional hazards regression analyses were applied to develop and validate the gene signature based on the TCGA dataset. The signature genes were then verified on GEPIA, Oncomine, and HPA platforms. Expression levels of corresponding genes were also measured by qRT-PCR and Western blotting in HBE, A549, and PC-9 cell lines. The prognostic signature based on eight genes (*TTK*, *HMMR*, *ASPM*, *CDCA8*, *KIF2C*, *CCNA2*, *CCNB2*, and *MKI67*) was established, which was independent of other clinical factors. The risk model offered better discrimination between risk groups, and patients with high-risk scores tended to have poor survival rate at 1-, 3- and 5-year follow-up. The model also presented better survival prediction in cancer-specific cohorts of age, gender, clinical stage III/IV, primary tumor 1/2, and lymph node metastasis 1/2. The signature genes, moreover, were highly expressed in A549 and PC-9 cells. In conclusion, the risk score signature could be used for prognostic estimation and as an independent risk factor for survival prediction in patients with LUAD.

## 1. Introduction

Lung cancer is still the deadliest cancer type worldwide. More than 1.6 million new patients are diagnosed each year with lung cancer, and the disease is associated with poor quality of life and high expenditure [1,2]. Lung adenocarcinoma (LUAD) is considered to be the most frequent type of lung cancer, and approximately 40% of lung cancer patients have LUAD [3]. Despite advances in cancer therapy in the past years, the prognosis of LUAD patients is still unsatisfactory, with poor overall survival (OS) rate. It is well known that the genetic alterations have critical functions on critical biological pathways during LUAD development and progression, and further promote the recurrence of cancer as well as lower the survival rate of patients [4,5]. Thus, it is necessary to identify reliable predictors for prognostic estimation, which could bring tremendously guiding value in the management of the LUAD. Especially, the construction of multiple-gene signature would benefit LUAD patients from prognostic prediction.

The widespread application of gene expression profiles and the availability of public genomic datasets enable researchers to study and elucidate the underlying molecular mechanisms of diseases at genome and transcriptome levels [6,7,8]. With the integration of genomics technology and bioinformatics analysis, gene microarrays have been widely applied to collect and study gene expression profiles in several cancer types [9,10]. While these tools allowed researchers to screen for tumor-associated genes and identify the core prognosis factors, a single gene biomarker cannot offer efficient survival prediction. A risk model based on multiple genes may offer a better prognostic factor for predicting the survival of patients [11,12]. In this study, we aimed to establish a multiple-gene risk score signature via systematic bioinformatics analysis.

Here, we obtained four gene expression profiles from the Gene Expression Omnibus (GEO) database (https://www.ncbi.nlm.nih.gov/geo/) and used these profiles to collect differentially expressed genes (DEGs) and to identify the signature genes. We then constructed protein-protein interaction (PPI) and subsequently performed a clustering module and biological pathways enrichment analyses. We adopted univariate and multivariate Cox proportional hazards regression analyses to construct the risk score model, and then used survival analyses as well as the receiver operating characteristic (ROC) curve to verify the sensitivity and specificity of this model. We also illustrate the prognostic value of the risk score signature. The detailed workflow of this work is provided in Figure 1.

## 2. Results

### 2.1. Identification of 573 DEGs Shared by Four GEO Profiles

To screen the DEGs between LUAD and normal lung tissues, we utilized the GSE18842, GSE75037, GSE101929, and GSE19188 gene expression profiles from the GEO database. The clinical characteristics of lung cancer patients contained in the Cancer Genome Atlas (TCGA) and these four GEO datasets are shown in Table 1. It is noteworthy that data on some clinical characteristics (particularly T classification, N classification, and M classification) were not provided for the four GEO datasets. The DEGs were screened in each of the four GEO datasets, respectively. First, we found 1840 DEGs in GSE18842, 2856 DEGs in GSE19188, 1959 DEGs in GSE75037, and 1458 DEGs in GSE101929, respectively. Of them, 812, 1313, 792, and 553 genes were upregulated and 1028, 1543, 1167, and 905 genes were downregulated in GSE18842, GSE19188, GSE 75037, and GSE101929 datasets, respectively (Figure 2A–D). Then, a total of 573 overlapping DEGs (197 upregulated and 376 downregulated)were screened among these four profiles (Figure 2E,F), which were used for the following analyses.These common DEGs significantly forcused on mitotic nuclear division (biological process, BP),extracellular matrix (cellular component, CC),and extracellular matrix structural constituent (molecular function, MF), as well as other functional pathways (Appendix A). Moreover, cytokine-cytokine receptor interaction, IL-17 signaling pathway, p53 signaling pathway and cell cycle were also main signaling pathways enriched by the 573 overlapping DEGs (Appendix A).

### 2.2. Candidate Core Genes Identification

STRING and Cytoscape 3.6.0 software programs were used to construct PPI network for the 573 overlapping DEGs and the network included 502 nodes and 4320 edges (Figure 3A). Moreover, we obtained four clustering modules from Cytoscape’s MCODE plug-in. Appendix A has shown that Cluster 1 included 72 nodes and 2326 edges with the highest MCODE score 65.521. Furthermore, it significantly focused on cell cycle, DNA replication, Mitotic M-M/G1 phases, M Phase, and other signaling pathways (adjusted *p* < 0.05) (Figure 3B). Meanwhile, other cluster modules were mainly enriched in cell surface interactions at the vascular wall, Hemostasis, Beta3 integrin cell surface interactions, and Peptide ligand-binding receptors (adjusted *p* < 0.05) (Figure 3C–E). More details are provided in Appendix A. In this study, clustering module 1 was considered as the key module. Then, the top 30 genes with largest degrees of connectivity in cluster 1 were shown in Table 2. We selected the first 20 genes with the highest degrees of connectivity as the candidate core genes for further study.

### 2.3. The Risk Model Based on the Eight Genes is Verified as an Independent Prognosis Factor

To establish the gene expression signature, we performed univariate and multivariate Cox regression analyses for the 20 candidate core genes and identified eight DEGs (*TTK*, *HMMR*, *ASPM*, *CDCA8*, *KIF2C*, *CCNB2*, *MKI67*, and *CCNA2*), as shown in Table 3. The risk score for OS was calculated as follows:

Risk score = ((−0.462) × *TTK* expression level) + (0.633 × *HMMT* expression level)

+ ((−0.550) × *ASPM* expression level) + ((−1.309) × *CDCA8* expression level)

+ (1.188 × *KIF2C* expression level) + ((−0.474) × *CCNB2* expression level)

+ (0.570 × *MKI67* expression level) + (0.484 × *CCNA2* expression level).

In the training group, LUAD patients were divided into high-risk and low-risk groups according to median risk score, which was used as cut-off value. The survival analysis revealed that the survival rate was distinctively lower in the high-risk group than low-risk group (*p* < 0.01), as shown in Figure 4A. Additionally, the risk model offered a survival prediction with AUCs: 0.706, 0.748, and 0.698 at 1-, 3- and 5-yearfollow-up, respectively (Figure 4B). The heatmap indicated that eight signature genes tended to have higher expression in high-risk patients (Figure 4C). The distribution of LUAD patients was plotted based on median cut-off value (Figure 4D). The survival status of patients in the training group was illustrated in Figure 4E. Meanwhile, we observed similar results in the testing group (Figure 5). Moreover, our risk model could be used as an independent prognosis predictor for LUAD patients in the training (*p* < 0.01) and testing (*p* < 0.001) groups (Table 4) in multivariate Cox analysis.

We further analyzed the TCGA dataset to evaluate the prognostic value of the risk score signature in LUAD patients (Figure 6). We found that the high-risk group was distinctively associated with worse OS in >65 age (*p* = 0.008), male (*p* = 0.023), stage III/IV (*p* = 0.008), N1-3 (*p* = 0.024), and T1-2 (*p* = 0.027) subgroups. Then, the ROC analysis verified the accuracy of our gene signature in OS prediction in subgroups with different clinical characteristics (Appendix A).The values of AUCs at 1, 3, and 5 years were between 0.965 and 0.610, which indicated that the risk model had a robust performance to predict OS probabilities for LUAD patients.


### 2.4. Signature Genes Show High Expression in LUAD Samples

The results from our correlation analysis showed that the risk score was positively associated with mRNA expression levels of all signature genes (*p* < 0.0001; Figure 7A), which was significantly higher in LUAD tissues compared with normal lung tissues (Figure 7B). We also found that *TTK*, *HMMR*, *CDCA8*, *CCNA2*, *CCNB2*, and *MKI67* proteins were over-expressed in LUAD tissues versus normal tissues (Appendix A). Compared to normal tissues, *ASPM* and *KIF2C* expressions were significantly elevated in lung cancer tissues within three datasets (Hou Lung, Selamat Lung, and Okayama Lung) [13,14,15] and five datasets (Hou Lung, Landi Lung, Okayama Lung, Stearman Lung, and Su Lung) [13,15,16,17,18], respectively (Appendix A).

### 2.5. Signature Genes with High Expression in LUAD Cells

Compared with human bronchial epithelial (HBE) cells, the mRNA levels of *TTK*, *HMMR*, *ASPM*, *CDCA8*, *KIF2C*, *CCNB2*, *MKI67*, and *CCNA2* were significantly upregulated in A549 and PC-9 cell lines(*p* < 0.0001, Figure 8A). Meanwhile, TTK, HMMR, ASPM, CDCA8, KIF2C, CCNB2, MKI67, and CCNA2 proteins were also significantly over-expressed in A549 and PC-9 compared to HBE cells(*p* < 0.05, Figure 8B, Appendix A).These results were consistent with expression levels of riskscore signature genes in LUAD tissues.

## 3. Discussion

Currently, LUAD is regarded as the most common subtype of lung cancer with a five-year OS rate between 4% and 17% [3]. The pathogenesis of LUAD involves a diverse and complex set of events, such as gene expression, autophagy activation, unexpected tumor microenvironment, immune cell infiltration, abnormal cell cycle, DNA methylation, epigenetic interactions, and other molecular and cellular events [19,20,21,22,23]. Moreover, accumulating evidence suggests that the corresponding signaling pathways, such as the MAPK, AKT-mTOR, and Wnt/β-catenin signaling pathways, are more frequently activated in LUAD [24,25,26]. While research into the tumor mechanism and treatment has made significant progress, the prognosis still remains poor in LUAD. Hence, it is urgent to identify the precise and effective prognostic signature to predict the survival of patients with LUAD.

Several risk score models based on gene expression for LUAD prognosis have been identified in the previous studies. The 9-gene signature (*HMMR*, *B4GALT1*, *SLC16A3*, *ANGPTL4*, *EXT1*, *GPC1*, *RBCK1*, *SOD1*, and *AGRN*) has been identified as an independent prognostic factor for OS in LUAD patients and is significantly associated with metastasis in the test series [2]. A 4-gene signature (*CTNNB1* or *β-catenin*, *SOX9*, *DVL3*, and *Wnt2b*) involved in Wnt/β-catenin pathway can significantly divide LUAD patients into high- and low-risk groups with different OS rates [27]. An immune signature consisting of 40 genes can effectively differentiate high- and low-risk groups among patients with stage I or II lung adenocarcinoma and independently predict OS [28]. Meanwhile, the clinical characteristics such as age and stage can improve the prognostic accuracy of the immune signature. A 6-gene signature (*RRAGB*, *RSPH9*, RPS6KL1, *RXFP1*, *RRM2*, and *RTL1*) can significantly stratify LUAD patients into high- and low-risk groups, and remain as an independent prognostic factor to estimate OS in a multivariate Cox proportional hazards model analysis [11]. An 8-gene signature (*DLGAP5*, *KIF11*, *RAD51AP1*, *CCNB1*, *AURKA*, *CDC6*, *OIP5*, and *NCAPG*) has been constructed by a multivariate Cox regression model that can clearly separate LUAD patients into groups with significantly different OS rates [29]. Our 8-gene prognostic signature (*TTK*, *HMMR*, *ASPM*, *CDCA8*, *KIF2C*, *MKI67*, *CCNA2*, and *CCNB2*) was different from these previously identified gene signatures, indicating it might be a novel tool for LUAD prognosis.

*TTK* is a mitotic checkpoint kinase that is highly expressed in several human cancers [30]. A high *TTK* expression, moreover, is positively associated with higher grade aggressiveness and therapeutic resistance in breast cancer, implicating the *TTK* might be involved in cancer cell proliferation and poor patient survival rate [31]. *HMMR* is also highly expressed in human multiple malignancies, such as gastric cancer and lung cancer, which may promote cancer progression and lower patient survival rate [2,32]. Ye et al. suggested that TGF-β signaling and Hippo pathway could contribute to sarcomagenesis and metastasis by upregulating the *HMMR* expression [33]. *ASPM*, the novel Wnt co-activator, augments the Wnt-β-catenin signaling to maintain the subpopulation of cancer stem cells in prostate cancer [34]. The expression of *ASPM* is incrementally upregulated in primary and metastatic lung cancer, indicating its potential roles in the occurrence and progression of lung cancer [35]. *CDCA8*, on the other hand, is involved in cell cycle regulation [36]. For example, *CDCA8* suppression leads to cell cycle G1 phase arrest, upregulations of p21 and p27 expressions, and the downregulations of CCND1 and Bcl-2 expressions [37]. Additionally, high *CDCA8* expression showed a significant correlation with poor survival in patients with cutaneous melanoma [38]. Results from integrated analyses show that *KIF2C* is overexpressed in several solid cancers and has already been identified as an important prognostic factor for cancers [39,40]. *KIF2C*, a mitotic centromere-associated kinase, acts on cellular senescence mainly via p53 signaling; it frequently induced T cell responses in certain cancers [39,41]. *CCNA2* has critical functions in controlling cell cycle at the G1/S and the G2/M transitions and is necessary for embryonic cell and hematopoietic lineage [42]. It has been reported that *CCNA2* might be associated with the progression of epithelial-mesenchymal transitions (EMT) and metastasis [43]. CCNA2 expression, moreover, is significantly upregulated in many cancer types according to the Human Protein Atlas database, implying its potential role in cancer development and progression. Similarly, *CCNB2* is also a key regulator of the cell cycle and may play a role in the development and progression of cancers in humans [44,45,46]. In this study, we established the risk score signature based on these eight genes and elucidated the prognostic value of the gene signature for LUAD.

Despite the encouraging predictive value in LUAD prognosis, our risk model still has some limitations. First, since our model is based on retrospective datasets and these datasets may have unbalanced clinical features with treatment heterogeneity, its efficacy needs to be validated in clinical trials with enough LUAD samples. Second, our risk model could not accurately predict survival possibilities for LUAD patients in <=65 age, female, Stage I and II, N0, and T3-4subgroups of LUAD patients, which might be due to limited samples. Meanwhile, the values of AUC at 1, 3, and 5 years were lower than 0.8, indicating that we should try to establish a new and powerful risk score model in the further analyses. Finally, the signaling pathways involved in the progression of LUAD need to be examined more thoroughly, which is beyond the scope of our current study.

## 4. Materials and Methods

### 4.1. Data Collection

Four gene expression profiles including GSE18842, GSE75037, GSE101929, and GSE19188 were obtained from the GEO online public database. GSE18842 contained 45 normal lung samples and 46 LUAD samples; GSE75037 contained 83 normal lung samples and 83 LUAD samples; GSE101929 contained 34 normal lung samples and 32 LUAD samples; and GSE19188 contained 65 normal lung samples and 91 LUAD samples.

### 4.2. Data Processing and Identification of DEGs

The normalization and log2 conversion for each GEO matrix data were performed by the limma package in R software (version 3.5.3). The limma package was applied to screen the DEGs in GSE18842, GSE75037, GSE19188, and GSE101929 profiles. The adjusted *p*-value < 0.05 and |log_2_FC| ≥ 1 were set as the screening criteria for DEGs.

### 4.3. PPI Network Construction and Clustering Module Analysis

The information regarding interactions of proteins was retrieved from the Search Tool for the Retrieval of Interacting Genes database (STRING) (https://string-db.org/), and a combined score of ≥0.4 was used as the cut-off criterion. Next, a protein-protein interaction (PPI) network was constructed using Cytoscape software (version 3.6.0) (https://cytoscape.org/). To glean specific biological information, molecularcomplex detection (MCODE), a Cytoscape plug-in, was used to cluster modules from the PPI network with the following parameters: degree cutoff = 2, Node Score cutoff = 0.2, and K-Core = 2. The enrichment analysis for the clustering modules was investigated with FunRich software (version 3.1.3).

### 4.4. Establishment and Validation of a Prognostic Signature

Patients in the Cancer Genome Atlas (TCGA) dataset were randomly divided into either a training group (228 LUAD cases) or a testing group (230 LUAD cases). After univariate Cox regression analysis, the DEGs that were significantly associated with OS were subjected to the least absolute shrinkage and selection operator (LASSO) regression analysis. The analysis was performed with the R “glmnet” package to screen out candidate prognostic genes in the training group. Subsequently, the candidate prognostic genes were included in the multivariate Cox regression analysis to establish the risk score formula, which could separate the high- and low-risk subgroups according to median risk score in patients with LUAD. Kaplan–Meier survival analysis and ROC curve were performed to determine the predictive value of the risk model. Univariate and multivariate Cox regression analyses were performed on the clinical data and risk score to evaluate whether the risk model was an independent prognosis factor from clinical parameters; the clinical relevance of the gene expression signature was also validated.

### 4.5. Validation of Corresponding Genes in Risk Score Signature

The Gene Expression Profiling Interactive Analysis (GEPIA) database (http://gepia.cancer-pku.cn) was adopted to illustrate the mRNA expression levels of signature genes in LUAD and normal lung tissues. The immunohistochemical results of corresponding genes from the Human Protein Atlas (HPA) (https://www.proteinatlas.org) were obtained to elucidate the protein expression levels of these genes. Furthermore, a meta-analysis was performed on the Oncomine database (https://www.oncomine.org) to identify the expression patterns of certain genes.

### 4.6. Cell Culture

HBE, A549, and PC-9 cells were grown in high glucose Dulbecco’s Modified Eagle’s media (DMEM; Hyclone, Logan, UT, USA) containing 10 % (*v*/*v*) fetal bovine serum (FBS; Gibco, Grand Island, NY, USA), and 1 % penicillin/streptomycin (MRC, Jintan, China) at 37 °C and 5% CO_2_. All cell lines were obtained from the Shanghai Cell Bank of Chinese Academy of Medical Sciences (Shanghai, China).

### 4.7. Quantitative Real-Time Polymerase Chain Reaction

Total RNA was extracted by using Total RNA Extraction Kit (Solarbo, Beijing, China), and reverse transcription was performed using the first-strand cDNA synthesis kit (Invitrogen, Carlsbad, CA, USA) following the manufacturers’ instructions. Real-time PCR was performed using Premix Ex Taq SYBR Green PCR (TaKaRa, Dalian, China) on an ABI PRISM 7300 (Applied Biosystems, Foster City, CA, USA) following the manufacturer’s guidelines. The primer sequences used are given in Appendix A.

### 4.8. Western Blot Analysis

After extraction of cell lysate, protein concentrations were quantified by the Bradford method (Beyotime, Shanghai, China) and proteins were separated on5–12% SDS-PAGE gels, which were then transferred to PVDF membranes (ThermoFisher, Waltham, MA, USA). The membranes were blocked with 5 % non-fat milk, incubated with primary antibody at 4 °C overnight, treated with horseradish peroxidase-conjugated secondary antibody (Bioss, Beijing, China) at room temperature for 1 h, and visualized on a Tanon 5200 (Tanon, Shanghai, China). Primary antibodies were purchased from Abcam (Cambridge, UK), and the following antibodies were used: TTK, HMMR, ASPM, CDCA8, KIF2C, CCNB2, MKI67, CCNA2, and β-actin. More information about primary antibodies was provided in Appendix A.

### 4.9. Statistical Analysis

All statistical analysis in this work was conducted using R software (version 3.6.0) and GraphPad Prism 7.0. The correlations between risk scores and expression levels of prognostic signature genes were analyzed via Pearson’s correlation test. Kaplan–Meier analysis was performed to estimate survival, and a log–rank test was used to compare between-group survival distributions. The predictive accuracy of our risk model was assessed by ROC analysis with the R “SurvivalROC” package. The univariate and multivariate Cox proportional hazards regression models were used to calculate regression coefficients and hazard ratios and to establish the risk score model. Two-tailed *p* < 0.05 was considered to be statistically significant.

## 5. Conclusions

We established a risk score signature for survival prediction in LUAD patients where patients with high-risk scores exhibited a poor survival rate. To some extent, our model may improve the prognostic accuracy in clinical application. Further studies are needed to improve the precision and reliability of the signature before our model can be implemented in clinical settings.

## Figures and Tables

**Figure 1 ijms-21-08479-f001:**
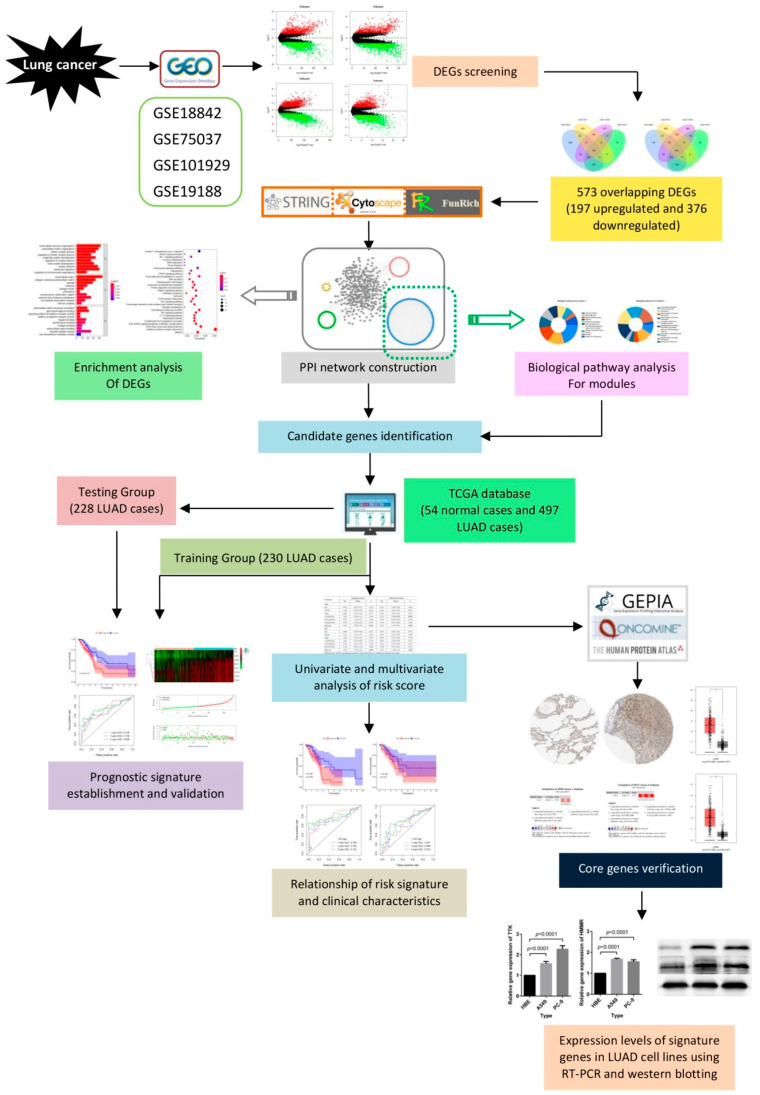
Flow chart for the construction of a risk score model in lung cancer.

**Figure 2 ijms-21-08479-f002:**
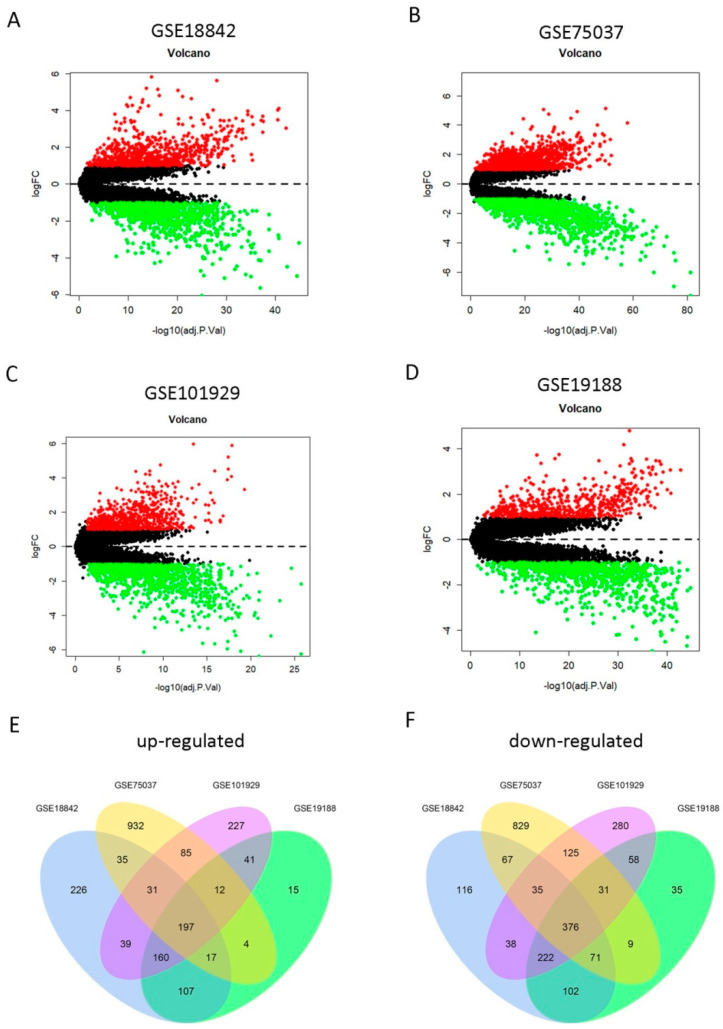
Identification of differentially expressed genes (DEGs) from four Gene Expression Omnibus (GEO) profiles. (**A**–**D**) Upregulated (redcolored spots) and downregulated (greencolored spots) DEGs in lung adenocarcinoma (LUAD) (compared to normal lung tissues) screened from the GEO profile GSE18842 (**A**), GSE75037 (**B**), GSE101929(**C**), and GSE19188 (**D**). (**E**–**F**) A total of 197 upregulated (**E**) and 376 downregulated (**F**) DEGs were shared among the four GEO expression profiles.

**Figure 3 ijms-21-08479-f003:**
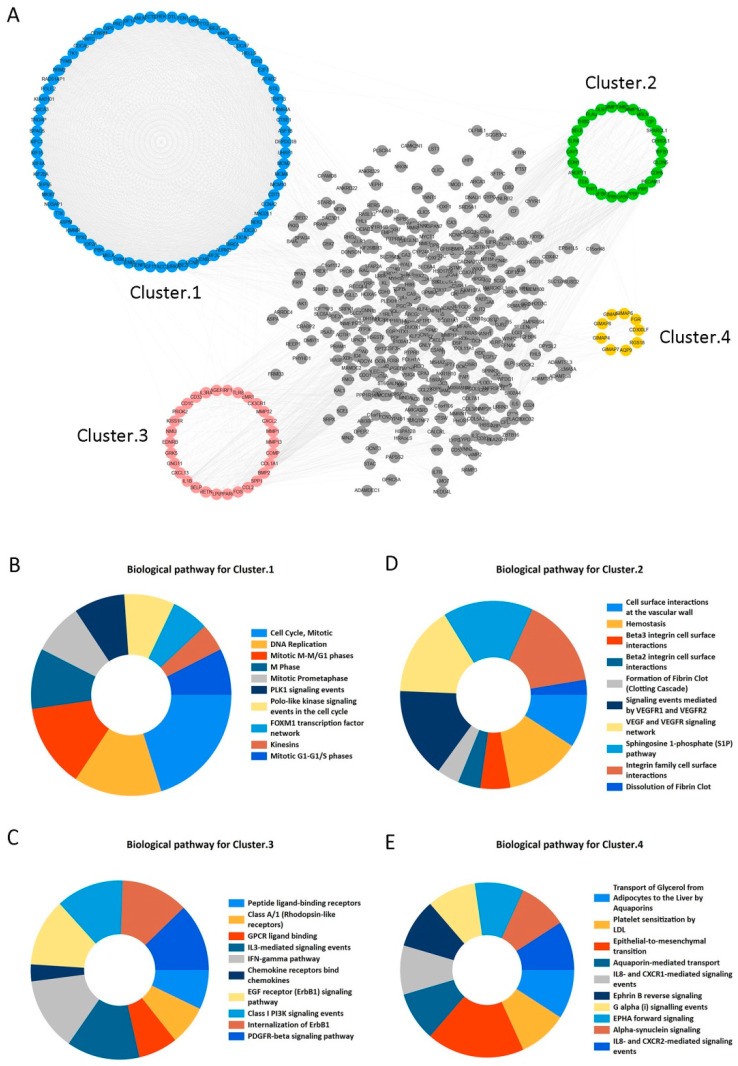
Protein-protein interaction (PPI) network construction and enrichment analysis for modules. (**A**) PPI network and module clustering analysis for the 573 shared DEGs. Blue, green, pink, and orange nodes represent the four clustering modules, of which cluster 1 (blue) had the highest MCODE score. (**B**–**E**) Biological pathway analysis for cluster 1 (**B**), cluster 2 (**C**), cluster 3 (**D**), and cluster 4 (**E**). Significant signaling pathways were mainly involved in cell cycle and DNA Replication.

**Figure 4 ijms-21-08479-f004:**
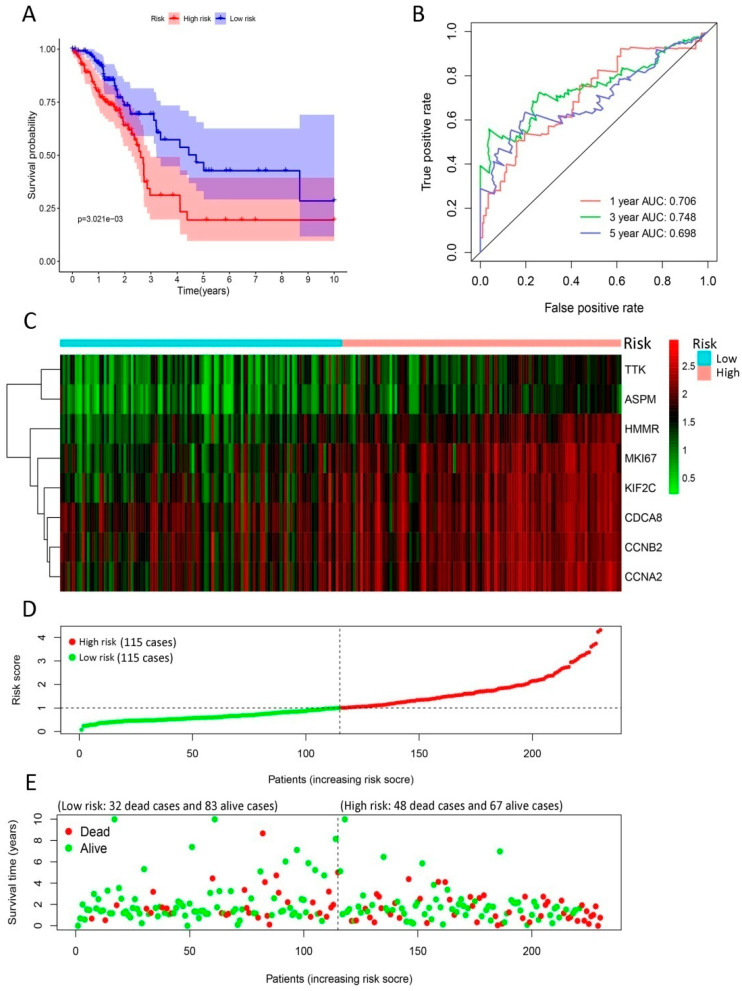
Development of gene signature in the training group. (**A**) Survival curves of high- and low-risk groups separated by gene signature. (**B**) Receiver operating characteristic (ROC) curves for survival risk predicted by gene signature for 1-, 3-, and 5-year follow-ups. (**C**–**E**) Expression heatmap of eight signature genes (**C**), risk score distribution (**D**), and survival status of patients (**E**). The risk score based on gene signature appears to be correlated with survival status for lung cancer patients in low- and high-risk groups.

**Figure 5 ijms-21-08479-f005:**
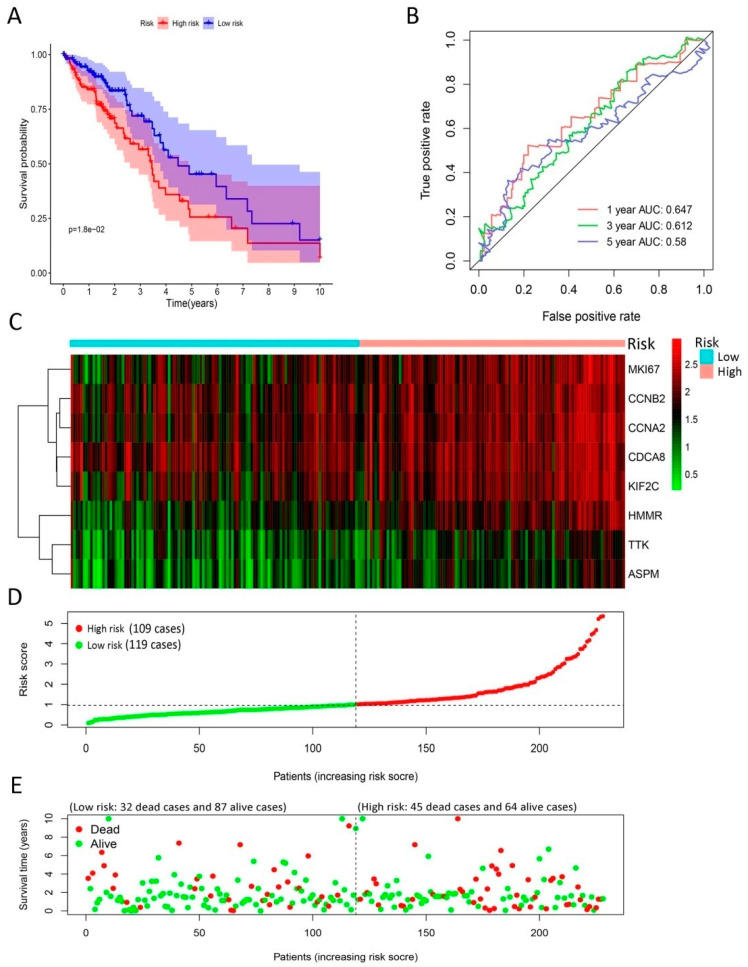
Validation of gene signature in the testing group. (**A**) Survival curve of high- and low-risk groups separated by gene signature. (**B**) ROC analysis for survival rate predicted by gene signaturefor 1-, 3-, and 5-year follow-ups. (**C**–**E**) Expression heatmap of eight signature genes (**C**), risk score distribution (**D**), and survival status of patients (**E**). The risk score based on gene signature in the testing group also appears to be correlated with survival status for lung cancer patients in low- and high-risk groups, which was consistent with that in training group.

**Figure 6 ijms-21-08479-f006:**
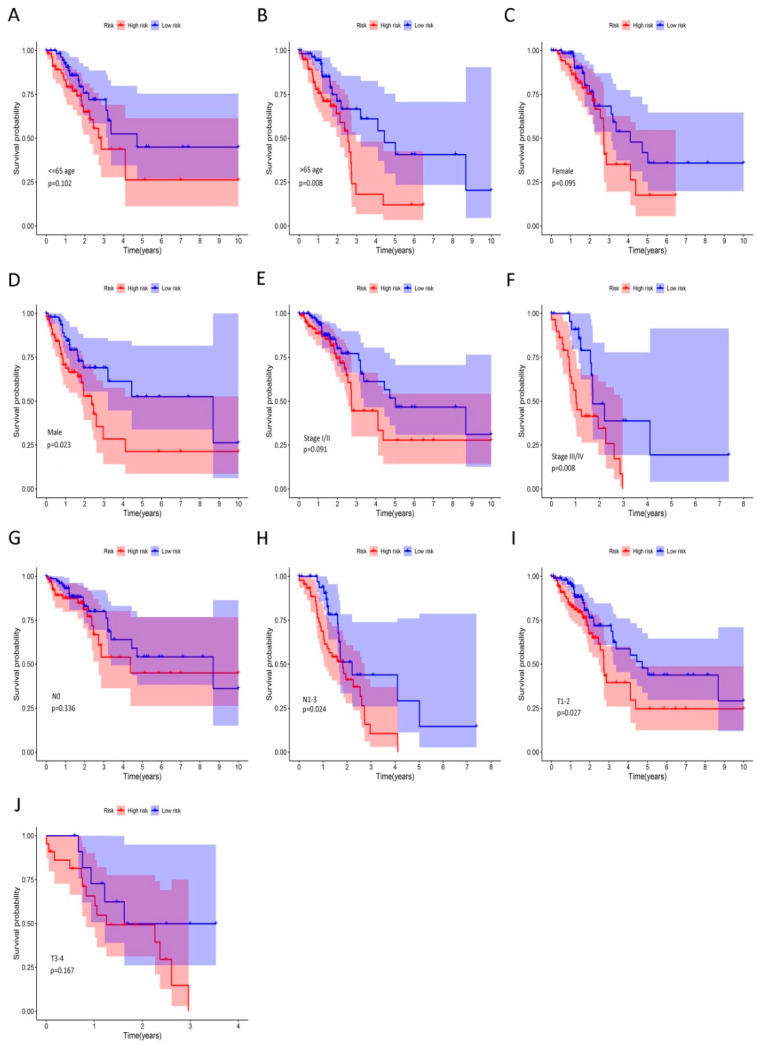
The prognostic value of riskscore signature in LUAD. (**A**–**J**) Kaplan–Meier curves for OS in patients with LUAD; clinical characteristics include age: ≤65 (**A**) and >65 (**B**), gender: female (**C**) and male (**D**), clinical stage I/II (**E**) and III/IV (**F**), lymph node metastasis: N0 (**G**) and N1-3 (**H**), and primary tumor: T1-2 (**I**) and T3-4 (**J**). Male, >65 age, stage III/IV, N1-2 and T1-2 subgroups were significantly associated with worse OS in high-risk group with respective *p* < 0.05.

**Figure 7 ijms-21-08479-f007:**
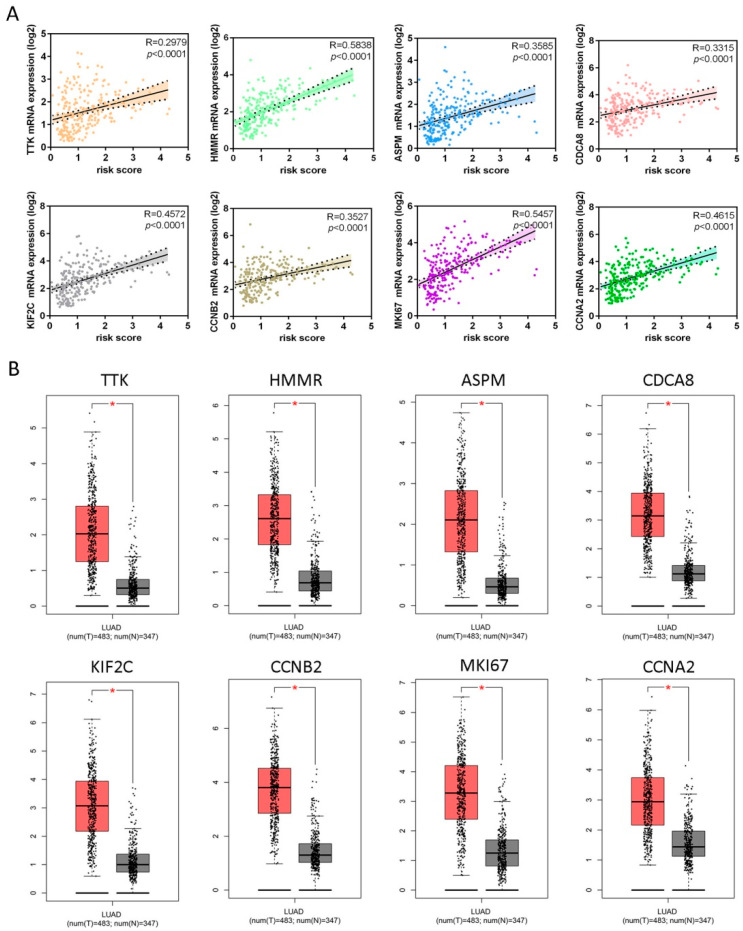
Validations of eight genes contained in the risk score model. (**A**) Scatter plots between mRNA expression of each gene (y-axis) and corresponding risk score (x-axis) in patients with LUAD for these eight genes, which showed significant positive Pearson correlation coefficients. (**B**) Comparisons of mRNA expression levels of each gene in LUAD tissues versus normal lung tissues, respectively. All genes had higher expression levels in cancer tissues than in normal lung tissues. The red and gray boxes represent cancer and normal tissues, respectively. The red-marked asterisk indicates that the differential expression of each mRNA is significant (*p* < 0.05).

**Figure 8 ijms-21-08479-f008:**
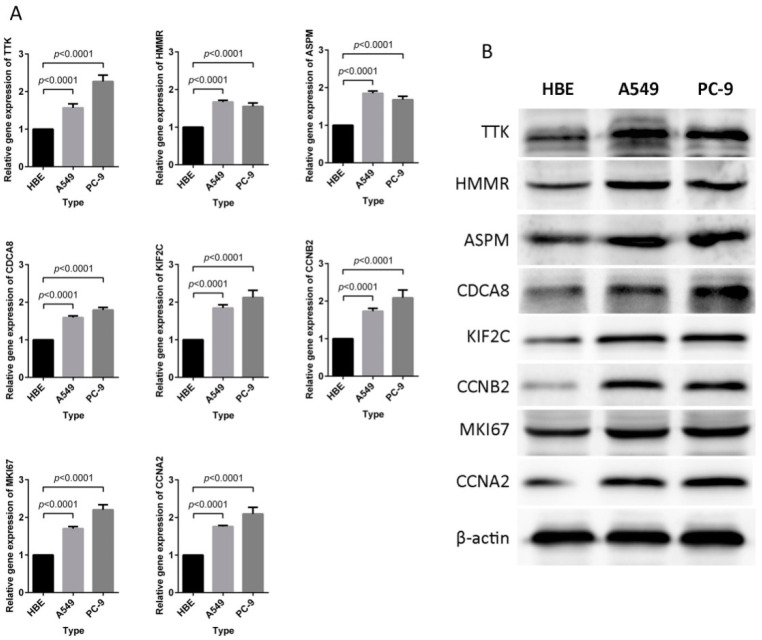
Expression profiles of eight signature genes in the HBE, A549, and PC-9 cell lines. (**A**) RT-PCR analysis of relative mRNA expression. The genes expression levels for each of the eight genes were significantly increased in A549 and PC-9 cells compared to HBE cells (*p* < 0.0001). (**B**) Western blot analysis of protein expression levels. Every protein appeared to have higher expression levels in A549 and PC-9 cells compared with HBE cells. The measurements of mRNA and protein expression levels were performed in triplicate.

**Table 1 ijms-21-08479-t001:** Clinical characteristics of patients with lung cancer in different datasets.

Characteristic	TCGA Data (n, %)	GSE18842 (n, %)	GSE75037 (n, %)	GSE101929 (n, %)	GSE19188 (n, %)	
**Platform**	Illumina HiSeq2000 RNA sequencing platform	Affymetrix Human Genome U133 Plus 2.0 Array	Illumina HumanWG-6 v3.0 expression beadchip	Affymetrix Human Genome U133 Plus 2.0 Array	Affymetrix Human Genome U133 Plus 2.0 Array	
**Samples**	**551 (100.0%)**	**91 (100.0%)**	**166 (100.0%)**	**66 (100.0%)**	**156 (100.0%)**	
Normal	54 (9.8%)	45 (49.5%)	83 (50.0%)	34 (51.5%)	65 (41.7%)	
Tumor	497 (90.2%)	46 (50.5%)	83 (50.0%)	32 (48.5%)	91 (58.3%)	
**Survival Status**	**486 (88.2%)**	NA	NA	**66 (100.0%)**	**82 (52.6%)**	
Death	162 (29.4%)	NA	NA	40 (60.6%)	50 (32.1%)	
Survival	324 (58.8%)	NA	NA	26 (39.4%)	32 (20.5%)	
**Age**	**467 (84.8%)**	NA	**166 (100.0%)**	**66 (100.0%)**	NA	
<=65	227 (41.2%)	NA	58 (34.9%)	43 (65.2%)	NA	
>65	240 (43.6%)	NA	108 (65.1%)	23 (34.8%)	NA	
**Gender**	**486 (88.2%)**	NA	**166 (100.0%)**	**66 (100.0%)**	**134 (85.9%)**	
Female	264 (47.9%)	NA	118 (71.1%)	38 (57.6%)	34 (21.8%)	
Male	222 (40.3%)	NA	48 (28.9%)	28 (42.4%)	100 (64.1%)	
**Stage**	**478 (86.8%)**	NA	**33 (19.9%)**	NA	NA	
I	262 (47.5%)	NA	20 (12.0%)	NA	NA	
II	112 (20.3%)	NA	8 (4.8%)	NA	NA	
III	79 (14.3%)	NA	5 (3.0%)	NA	NA	
IV	25 (4.5%)	NA	NA	NA	NA	
**T classification**	**483 (87.7%)**	NA	NA	NA	NA	
T1	163 (29.6%)	NA	NA	NA	NA	
T2	260 (47.2%)	NA	NA	NA	NA	
T3	41 (7.4%)	NA	NA	NA	NA	
T4	19 (3.4%)	NA	NA	NA	NA	
**N classification**	**474 (86.0%)**	NA	NA	NA	NA	
N0	312 (56.6%)	NA	NA	NA	NA	
N1	90 (16.3%)	NA	NA	NA	NA	
N2	70 (12.7%)	NA	NA	NA	NA	
N3	2 (0.4%)	NA	NA	NA	NA	
**M classification**	**357 (64.8%)**	NA	NA	NA	NA	
M0	333 (60.4%)	NA	NA	NA	NA	
M1	24 (4.4%)	NA	NA	NA	NA	

For TNM classification, T, N, and M refer to primary tumor, regional lymph nodes, and distant metastasis, respectively. Abbreviations: TCGA, The Cancer Genome Atlas; NA, not available.

**Table 2 ijms-21-08479-t002:** Top 30 genes with highest degrees of connectivity in clustering module 1.

NO.	Gene	Degree	NO.	Gene	Degree	NO.	Gene	Degree
1	*UBE2C*	71	11	*ASPM*	71	21	*KIAA0101*	70
2	*NUSAP1*	71	12	*CENPF*	71	22	*SPAG5*	70
3	*TPX2*	71	13	*CDCA8*	71	23	*KIF15*	70
4	*PBK*	71	14	*KIF2C*	71	24	*CEP55*	70
5	*MELK*	71	15	*AURKB*	71	25	*CENPE*	70
6	*TTK*	71	16	*CCNB2*	71	26	*CDC20*	70
7	*KIF11*	71	17	*KIF20A*	71	27	*BIRC5*	70
8	*TOP2A*	71	18	*MKI67*	71	28	*MCM10*	70
9	*HMMR*	71	19	*CCNA2*	71	29	*MAD2L1*	70
10	*RRM2*	71	20	*CCNB1*	71	30	*AURKA*	70

**Table 3 ijms-21-08479-t003:** Identification of gene expression signature by univariate and multivariate Cox regression analysis.

NO.	Gene	Univariate Analysis *			Multivariate Analysis **		
		HR	95%CI	*p*	HR	95%CI	Coef.
1	*UBE2C*	1.145	1.033–1.270	0.010	---	---	---
2	*TPX2*	1.226	1.089–1.381	0.001	---	---	---
3	*PBK*	1.264	1.100–1.453	0.001	---	---	---
4	*MELK*	1.233	1.069–1.422	0.004	---	---	---
5	*TTK*	1.247	1.053–1.477	0.010	0.630	0.341–1.165	−0.462
6	*KIF11*	1.358	1.148–1.608	<0.001	---	---	---
7	*TOP2A*	1.178	1.043–1.331	0.008	---	---	---
8	*HMMR*	1.472	1.243–1.742	<0.001	1.883	1.153–3.074	0.633
9	*RRM2*	1.298	1.128–1.493	<0.001	---	---	---
10	*ASPM*	1.409	1.169–1.698	<0.001	0.577	0.287–1.159	−0.550
11	*CENPF*	1.293	1.112–1.503	0.001	---	---	---
12	*CDCA8*	1.206	1.039–1.401	0.014	0.270	0.100–0.730	−1.309
13	*KIF2C*	1.234	1.074–1.417	0.003	3.281	1.232–8.738	1.188
14	*AURKB*	1.188	1.039–1.358	0.012	---	---	---
15	*CCNB2*	1.258	1.085–1.458	0.002	0.622	0.329–1.178	−0.474
16	*KIF20A*	1.350	1.136–1.605	0.001	---	---	---
17	*MKI67*	1.309	1.129–1.518	<0.001	1.768	1.103–2.835	0.570
18	*CCNA2*	1.328	1.150–1.533	<0.001	1.622	0.889–2.959	0.484
19	*CCNB1*	1.321	1.136–1.535	<0.001	---	---	---
20	*NUSAP1*	1.293	1.107–1.511	0.001	---	---	---

* The 20 DEGs were significantly associated with overall survival (*p* < 0.05) using univariate Cox regression analysis. ** Then, a least absolute shrinkage and selection operator (LASSO) regression on these 20 DEGs was performed to identify the most informative gene set for survival prediction. Finally, eight genes marked in gray in the table were selected to perform multivariate Cox regression analysis and to generate a prognostic risk model according to their respective regression coefficients. Abbreviations: HR, hazard ratio; CI, confidence interval; *p*, *p*-value; Coef., coefficient.

**Table 4 ijms-21-08479-t004:** Univariate and multivariate Cox regression analysis for risk score on overall survival of patients with LUAD.

Parameter	Univariate Analysis			Multivariate Analysis		
	HR	95%CI	*p*	HR	95%CI	*p*
**Training Group**						
Age	0.492	0.169–1.431	0.193	0.612	0.204–1.838	0.381
Gender	1.128	0.656–1.940	0.663	1.062	0.605–1.865	0.833
Stage	0.808	0.421–1.554	0.523	0.295	0.061–1.412	0.126
T classification	1.753	0.826–3.721	0.144	2.755	1.086–6.988	**0.033**
M classification	0.961	0.233–3.97	0.956	3.162	0.328–30.501	0.320
N classification	0.914	0.578–1.444	0.699	1.534	0.637–3.690	0.340
RiskScore	3.285	1.681–6.420	**0.001**	2.931	1.474–5.829	**0.002**
**Testing group**						
Age	1.008	0.978–1.038	0.612	0.991	0.962–1.021	0.556
Gender	0.623	0.352–1.105	0.106	0.595	0.327–1.083	0.089
Stage	1.039	0.792–1.363	0.782	0.861	0.371–2.001	0.728
T classification	1.063	0.755–1.496	0.726	1.083	0.704–1.664	0.717
M classification	0.943	0.372–2.393	0.902	0.916	0.142–5.909	0.926
N classification	1.178	0.802–1.730	0.404	1.628	0.751–3.529	0.217
RiskScore	1.594	1.256–2.022	**<0.001**	1.662	1.284–2.152	**<0.001**

For TNM classification, T, N, and M refer to primary tumor, regional lymph nodes, and distant metastasis, respectively. Abbreviations: LUAD, lung adenocarcinoma; HR, hazard ratio; CI, confidence interval; *p*, *p*-value.

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
