# Peer review of "Establishment of a Gene Signature to Predict Prognosis for Patients with Lung Adenocarcinoma"

_ijms, 2020, doi:10.3390/ijms21228479_

Round 1
Reviewer 1 Report
In this manuscript, based on their virtual analysis of GSE and TCGA data, the authors report the development of a transcriptional signature of eight genes that may potentially represent a prognostic factor for lung adenocarcinoma (LUA) patients. Considering hundreds of genes that may be differentially expressed in LUA tissues compared to normal tissue, the rationale and criteria for these selected genes from these top 30 genes aren’t clear enough and some stochastic; the necessity, clinical relevance, practicality, and novelty of the 8-gene transcriptional signature are very limited compared to these available prognostic biomarkers. Moreover, the proposed signature isn’t further validated and explored by using their own samples and additional experiments; and it doesn't provide important insight into the mechanisms and significance of these genes in LUA carcinogenesis and progression. In addition, the presentation of the manuscript is not concise enough, and many data are very preliminary, for example Figure 1/2 doesn’t include essential information.
Author Response
Dear reviewer,
RE: Manuscript ID ijms-930056
We would like to thank International Journal of Molecular Sciences for giving us the opportunity to revise our manuscript. And we thank you so much for your careful read and thoughtful comments on previous draft. We have carefully taken your comments into consideration in preparing our revision, which has resulted in a paper that is clearer, more compelling, and broader.
Below is our response to your comments. Thanks for all the help.
Best wishes,
Fan Li
Corresponding Author
Revision — authors’ response
- In response to the reviewer’s requests, we have clearly explained the rationale and criteria for candidate core genes in the text. Additionally, we added a new supplementary table (Supplementary Table S1) to support our findings. The following summarizes how we selected the candidate core genes. Firstly, we identified 573 overlapping DEGs from four GEO profiles (GSE18842, GSE75037, GSE19188, and GSE101929), and then we have performed the protein-protein interaction (PPI) network analysis for the 573 common DEGs using the Search Tool for the Retrieval of Interacting Genes database (STRING) with a combined score of ≥0.4. Next, Cytoscape software (version 3.6.0) and a Cytoscape plug-in (MCODE) were used to construct the PPI network and perform clustering module analysis, of which cluster 1 was considered as our key module with the highest MCODE score (65.521). Meanwhile, we calculated the degrees of connectivity of each gene in cluster 1, and Table 2 presented the top 30 genes with high connectivity. Finally, we only selected the first 20 genes with the highest degrees of connectivity (71) as the candidate core genes for the following analyses.
- As suggested by the reviewer, we have made additional changes in the Introduction and Discussion in hopes that these will support our findings. In this manuscript, we focused on identifying reliable predictors for prognostic estimation, especially multiple-gene signature, which could bring tremendously guiding value in the management of the LUAD. Although there were some limitations for the 8-gene signature when compared to available prognostic biomarkers, our risk score model was different from previous gene signatures, indicating it might be a novel tool for LUAD prognosis. To some extent, our model may improve the prognostic accuracy in clinical application. Meanwhile, we have to acknowledge that further studies are needed to improve the precision and reliability of the signature before our model can be implemented in clinical settings.
- In the study, the 8-gene signature was developed based on bioinformatics analyses, and each gene was validated at mRNA and protein levels using RT-PCR and western blotting in A549, PC-9, and HBE cell lines. Additionally, the representative IHC pictures of TTK, HMMR, CDCA8, CCNA2, CCNB2, and MKI67 proteins have been retrieved from HPA database. Meanwhile, the meta-analysis of ASPM and KIF2C was conducted according to Oncomine database. Both of them, to some extent, could accurately depict the expression trends of the 8 genes. In the other hand, additional experiments in clinical samples will be conducted in the further studies.
- As requested by the reviewer, we have accurately modified the manuscript and added additional data in order to turn the paper into a significantly stronger manuscript. Fig.1 functioned as a guideline, which would help readers to catch our main analysis ideas and methods in the paper. Fig.2 mainly presented the DEGs distribution in four GEO profiles (GSE18842, GSE19188, GSE 75037, and GSE101929). Meanwhile, we could identify 573 overlapping DEGs among the four GEO profiles including 197 up-regulated and 376 down-regulated genes, which were used for following analyses.
Reviewer 2 Report
Using different databases and bioinformatic tools, the authors identified a risk score signature of eight genes that can predict prognosis in lung adenocarcinoma (LUAD). The study made use of four gene expression profiles downloaded from GEO online public database, protein interaction networks defined by the STRING database and Cytoscape software, clustering module analysis and validation of the signature via COX proportional hazard regression analyses on the TCGA dataset as well as further corroboration of corresponding genes’ expression using the Gene Expression Profiling Interactive Analysis, Oncomine, and Human Protein Atlas platforms. Finally, the eight genes’ expression levels were investigated by qRT-PCR and Western blotting in LUAD cell lines. The study is well-designed, but the manuscript suffers from lack of originality (many papers have been published on prognostic gene signatures for LUAD) and insufficient description of the methods and especially of the results. A substantial revision of these sections and additional focus on the clinical implications of the gene signature are necessary before the manuscript is publishable.
SPECIFIC POINTS
Line 38-39, “Lung cancer is one of the human malignant neoplasms that cause the highest cancer-related mortalities worldwide”: actually, lung cancer is still the deadliest cancer type worldwide.
Line 40, “LUAD is considered to be the most frequent malignant lung cancer”: Please eliminate “malignant” and replace it with "the most frequent type of lung cancer" or similar, as a cancer by definition is a malignant disease.
Line 45-48, “As the single gene biomarker or prognosis system within the same tumor, node, and metastases (TNM) cannot accurately predict the prognostic factors in patients with LUAD, it is therefore important to develop a reliable risk score signature for survival prediction in LUAD”: the sentence is difficult to understand and should be formulated more properly in English. Moreover, it should be explained why the TNM system does not predict prognosis, when it is still routinely utilized in the clinics.
Methods, line 301-302: reformulate the sentence as proteins and not protein concentrations were separated om SDS-PAGE.
For the sake of reproducibility, specify the concentrations of each Ab used and whether they were polyclonal Abs or monoclonal Abs and from which species.
Table 1 should describe better the demographic and clinical characteristics of the patients in the different datasets. For instance, survival data and age data are missing for two GEO datasets, likewise one GEO data set is without gender data and three GEO data sets are without stage data, while TNM data are only shown for the TCGA cohort. It should be explained both in the table (for ex. use NA and a table legend instead of empty columns) and the text, why all these data are missing. Moreover, it would be relevant for the study to indicate what kind of treatment, if any, the patients in the different data sets have received. Indeed, as also mentioned by the authors in the Discussion, treatment heterogeneity can influence the prognostic impact of gene signatures.
Fig. 3A: Ckyster.3 should be Cluster.3. Moreover, the name of genes in the 4 clusters are very small, it would be difficult to read them in a printed article.
Results, paragraph 2.3 and table 3: It is not clear how the eight DEGs of the final gene signature, which are marked in grey in Table 3, were selected. Please explain more extensively specifying the criteria for the selection and why the 8 genes were selected instead of the other 12 DEGs.
After screening the 4 GEO data sets, the authors identified 573 overlapping DEGs, including 197 upregulated and 376 downregulated genes. However, it is not clear whether they used both upregulated and downregulated DEGs for the following analyses and for the selection of the eight-gene-signature. The authors should elaborate on that in general terms in the manuscript.
In this respect, the risk score defined on line 108-111 needs to be explained more clearly, particularly what -0.462, 0.663, -0.550, -1.309, 1.188, -0.474, 0.570 and 0.484 represent in the calculation of the risk score and whether any of the 8 genes were upregulated or downregulated. The impression one gets by reading the whole manuscript is that only overexpressed genes are represented in the final prognostic gene signature, but this needs to be stated clearly and the reasons for this choice likewise clarified. Define also better the high-risk and low-risk groups.
Line 113-115, “Additionally, the risk model offered an acceptable survival prediction with AUCs: 0.706, 0.748, and 0.698 at 1-, 3- and 5-yearfollow-up, respectively”. The authors appropriately call the survival prediction “acceptable” in the training group, but to be clinically relevant one would expect AUC > 0.8, which is not the case. The values of AUC at 1, 3 and 5 years are even lower (between 0.647 and 0.58) for the testing set in fig 5, which questions the prognostic strength of the gene signature. An explanation or comment regarding these concerns is needed in the manuscript .
Fig. 4C and 5C: To be consistent with other figures, write high risk and low risk. Fig. 4D-E and 5D-E: on the X axis “socre” should be “score”. Fig. 4E and 5E: somehow the authors should indicate whether the differences in survival and death are significant.
Line 125-126, “The ROC analysis further verified the survival prediction of our gene signature in the subgroups with different clinical characteristics”: In supplementary Fig. S2, the ROC curve for stage I-II has better AUC values than stage III-IV contradicting a bit the statement with respect to Fig. 6, in which the difference for stage III-IV is significant but that for stage I-II is not. How do the authors explain this apparent discrepancy?
Line 153-156, “significantly elevated in lung cancer tissues compared with that in normal tissues in three datasets (Hou Lung, Garber Lung, and Okayama Lung) and five datasets (Hou Lung, Landi Lung, Okayama Lung, Stearman Lung, and Su Lung), respectively”: the corresponding references should be cited here too and not only in the supplementary Fig. S3B.
Supplementary Fig. S3A: for completeness, please provide also a representative picture for the upregulation of ASPM and KIF2C proteins in LUAD vs. non-neoplastic tissue, as done for the other 6 genes in the figure.
Figure 7: The yellow-marked asterisk presumably indicates that the differential expression of each mRNA is significant. This should be explained in the figure legend.
Figure 8: Please explain more extensively the nomogram, as it is difficult to grasp right away (upregulated and downregulated genes, points, values for 1-yr, 3-yr and 5-yr survival).
Figure 9 and signature genes with high expression in LUAD cells: Were the measurements of mRNA expression shown in fig. 9A performed in triplicate? Please specify. Also describe the western blot in the Results, right now it is only briefly mentioned in the figure legend. Also, a figure with quantification of protein expression in the cell lines should be shown.
Discussion, line 183-184, “progress, the diagnosis and prognosis remain poor in LUAD”: what do the authors mean with diagnosis remaining poor? It should be formulated more clearly. The prognosis of LUAD, indeed, remains poor, but a "poor diagnosis" means not well performed. Is that the meaning of the sentence? The manuscript does not deal directly with LUAD diagnosis therefore the sentence is confusing.
By the same token, line 190-191, “ In this study, we utilized these tools to identify gene expression signature involved in the diagnosis and prognosis of LUAD”: it is unclear how the gene expression signature, which is prognostic, is involved in diagnosis. The concept should be discussed more clearly.
Line 198-199, “repression of the p21 expression, the checkpoint of the cell cycle”: p21 is not "the checkpoint of the cell cycle" but a protein participating in the G1/S and G2/M check-points of the cell cycle by binding and inhibiting the activity of cyclin-CDK4/6, -CDK2, and -CDK1 complexes. Its main function is to reduce cell cycle progression. Furthermore, it should be elucidated why p21 is mentioned in the Discussion as example.
Line 202-203: the sentence “furthermore, demonstrated that FOXM1 attributed to the LUAD growth and migration by co-expressing with CENE and regulating MMP2 expression” should be rephrased in a more understandable manner.
Line 242-243: likewise, the sentence “Second, Our model did not establish a better correlation with overall survival in several subgroups of patients with LUAD due to limited samples in subgroups” should be formulated more clearly.
The authors ought to emphasize further in the Discussion the clinical value of the risk model. Moreover, they should discuss their data with respect to other published prognostic gene signatures.
As minor point, some typos should be corrected, for instance: line 51: “spliceosome alternations” should be spliceosome alterations, line 74: are provided; line 115: 5-year follow-up; line 262: molecular complex detection.
Author Response
Dear reviewer,
RE: Manuscript ID ijms-930056
We would like to thank International Journal of Molecular Sciences for giving us the opportunity to revise our manuscript. And we thank you so much for your careful read and thoughtful comments on previous draft. We have carefully taken your comments into consideration in preparing our revision, which has resulted in a paper that is clearer, more compelling, and broader.
Below is our response to your comments. Thanks for all the help.
Best wishes,
Fan Li
Corresponding Author
Revision — authors’ response
Specific points:
- As suggested by the reviewer, we have modified the sentence of “Lung cancer is one of the human malignant neoplasms that cause the highest cancer-related mortalities worldwide” on line 37.
- As suggested by the reviewer, we have replaced “malignant” with "the most frequent type of lung cancer" on line 39-40.
- As suggested by the reviewer, we have additional changes in response to “As the single gene biomarker or prognosis system within the same tumor, node, and metastases (TNM) cannot accurately predict the prognostic factors in patients with LUAD, it is therefore important to develop a reliable risk score signature for survival prediction in LUAD” in the introduction
- As suggested by the reviewer, we have made an accurate modification for the method “4.8 western blot analysis”. Meanwhile, we have provided additional information about primary antibodies in Supplementary Table S4.
- As suggested by the reviewer, we have made an explanation for the missing data both in the table 1 and the text. And we have used NA and a table legend instead of empty columns. Additionally, the four GEO profiles (GSE18842, GSE75037, GSE19188, and GSE101929) and TCGA datasets didn’t provide the information about what kind of treatment the patients have received.
- As suggested by the reviewer, we have corrected the typescript error in Fig.3A. Additionally, we have added new data to address the problem of small font in the 4 clusters. The new data were depicted in Supplementary Table S1. As lots of genes in the PPI network, enlarging font still can’t clearly present the gene symbols in the network. So, we provided a supplementary table including gene symbols in the 4 clusters.
- As suggested by the reviewer, we have extensively explained how we selected the 8 genes as our risk score model both in the results of paragraph 2.3 and the legend of table 3.
- As suggested by the reviewer, we have elaborated that the 573 overlapping DEGs were used for following analyses in the results of paragraph 2.1. Meanwhile, the results of paragraph 2.2 and paragraph 2.3 have depicted the detailed processes and methods for the selection of 8-gene signature.
The following summarizes how we selected the 8-gene signature. Firstly, we identified 573 overlapping DEGs from four GEO profiles (GSE18842, GSE75037, GSE19188, and GSE101929), and then we have performed the protein-protein interaction (PPI) network analysis for the 573 common DEGs using the Search Tool for the Retrieval of Interacting Genes database (STRING) with a combined score of ≥0.4. Next, Cytoscape software (version 3.6.0) and a Cytoscape plug-in (MCODE) were used to construct the PPI network and perform clustering module analysis, of which cluster 1 was considered as our key module with the highest MCODE score (65.521). Meanwhile, we calculated the degrees of connectivity of each gene in cluster 1, and Table 2 presented the top 30 genes with high connectivity. Finally, we only selected the first 20 genes with the highest degrees of connectivity (degree=71) as the candidate core genes, which were used for the selection of 8-gene signature.
- As suggested by the reviewer, we have clearly explained what 0.462, 0.663, -0.550, -1.309, 1.188, -0.474, 0.570 and 0.484 represent in the calculation of the risk score in the legend of Table 3. Meanwhile, the expression level of each gene in our risk model was up-regulated in LUAD versus normal lung tissues. But, that didn’t mean all genes in other risk models were over-expressed in cancer tissues. The 8-gene signature was developed by bioinformatics analyses as the results of paragraph 2.3 in the study. Additionally, we have clearly defined the high- and low-risk groups in the results of paragraph 2.3.
- As suggested by the reviewer, we have corrected the typescript errors in Fig. 4D-E and 5D-E. Meanwhile, we have re-depicted the results of Fig. 4E and 5E in the results of paragraph 2.3.
- The AUC values were used to estimate the accuracy of the risk score model in overall survival prediction. The higher the value, the higher the accuracy. In Fig.6, the risk model could significantly predict the overall survival for patients with stage III&IV (p=0.008), but not patients with stage I&II (p=0.091). The ROC curve for stage I&II has better AUC values than stage III&IV in supplementary Fig.S2, indicating that the accuracy of the risk model in overall survival probability for patients with stage I&II was higher than those patients with stage III&IV. And it implied that the risk model could not be used for overall survival prediction of patients with stage I&II. There was no contradicting between figures.
- As suggested by the reviewer, we have cited the corresponding references in the text in response to the meta-analysis for ASPM and KIF2C expression.
- The representative IHC pictures of TTK, HMMR, CDCA8, CCNA2, CCNB2, and MKI67 proteins were retrieved from HPA database, but the database didn’t provide IHC pictures for the ASPM and KIF2C proteins. In the other hand, we performed the meta-analysis of ASPM and KIF2C to make up for the deficiency, which also depicted the expression trends of ASPM and KIF2C.
- As suggested by the reviewer, we have depicted the meaning of the yellow-marked asterisk in the legend of Fig.7.
- As suggested by the reviewer, we have more extensively explained the nomogram in the legend of Fig.8.
- The measurements of mRNA and protein expression shown in fig. 9A were performed in triplicate, which was specified in the Fig.9 legend. Meanwhile, the quantification of protein expression in the cell lines has been shown in supplementary Fig. S4.
- In the study, we focused on the prognosis of LUAD but not diagnosis of LUAD. And we didn’t pay more attention to the description in the original manuscript. Now, we have corrected the inaccurate and confusing sentences in the paper according to the suggestions of the reviewer.
- As suggested by the reviewer, we have clearly re-formulated the sentence of “Second, Our model did not establish a better correlation with overall survival in several subgroups of patients with LUAD due to limited samples in subgroups” on line 502-506.
- As suggested by the reviewer, we have made additional changes in the Discussion. And we further emphasized the clinical value of the risk model with respect to other gene signatures.
Minor points:
We have made modifications for the minor points accordingly in the revised manuscript.
Reviewer 3 Report
My main comments have to do with:
1) Language use. Some sentences could have been re-written in a more easy-to-read way.
2) In addition, it seems that the results are not adequately presented, apart from mentioning the corresponding Figures. The figure legends also, do not describe the main findings on each part.
3) Gene Ontology enrichment analysis of the 8 genes on the signature should be an integral part of the biological pathway analysis part, not in Suppl. data. Also, please describe GO enrichment findings, accordingly.
Minor comments:
Please correct the typo "Ckyster 3" in Fig. 3A
Please explain differences or commonalities in the main biological pathways found across clusters 1-3 in Fig. 3B-E.
Please explain in the legend of Figure 8 what are the main results of the nomogram.
Author Response
Dear reviewer,
RE: Manuscript ID ijms-930056
We would like to thank International Journal of Molecular Sciences for giving us the opportunity to revise our manuscript. And we thank you so much for your careful read and thoughtful comments on previous draft. We have carefully taken your comments into consideration in preparing our revision, which has resulted in a paper that is clearer, more compelling, and broader.
Below is our response to your comments. Thanks for all the help.
Best wishes,
Fan Li
Corresponding Author
Revision — authors’ response
Major comments:
- As suggested by the reviewer, we have re-written some paradoxical sentences in a more easy-to-read way.
- As requested by the reviewer, we have re-formulated our results in a more accurate way to support our findings. Meanwhile, the main findings on each part have been adequately presented in corresponding figure legends.
- In response to the reviewer’s suggestion, we didn’t perform the GO enrichment analysis for the 8 signature genes, while we conducted the GO enrichment analysis for the 573 overlapping DEGs in the study. We have re-explained the results on the part because of our previously ambiguousexpression. Additionally, we have added new descriptions for the GO enrichment analysis in the text.
Minor comments:
- We have corrected the typescript error in Fig. 3A.
- We have explained the differences or commonalities in the main biological pathways found in Fig. 3B-E.
- We have explained the main results of the nomogram in the legend of Fig. 8.
Round 2
Reviewer 2 Report
The Authors’ point-by-point reply to the initial comments is a bit confusing (not so accurate). However, reading through the revised MS shows that they have essentially addressed most of the points of criticisms and made most of the suggested adjustments. The MS is therefore worth publication.
It can be additionally suggested to remove figure 8 (nomogram) from the MS, as it remains poorly explained and does not seem to add much to the results and conclusions of the paper anyway (the figure and the corresponding two lines in the text can be removed at the galley proof stage).
Author Response
Dear reviewer,
RE: Manuscript ID ijms-930056
We sincerely express our thanks to you once again for your careful read and thoughtful comments on our manuscript. We have carefully taken your comments into consideration in preparing our revision. Below is our response to your comments. Thanks for all the help.
Best wishes,
Fan Li
Corresponding Author
Revision — authors’ response
Minor points:
As suggested by the reviewer, we have removed figure 8 (nomogram) and the corresponding two lines (line 177-179) from the MS.
